# Feasibility Study on the Development of a Deployable Tactical EMP Tent for a Sustainable Military Facility

**Kukjoo Kim** [1,2], **Kyung-Ryeung Min** [3] **and Young-Jun Park** [1,2,*]

1   Department of Civil Engineering and Environmental Sciences, Korea Military Academy, Seoul 01805, Korea; klauskim@ufl.edu
2   Nuclear WMD Protection Research Center, Korea Military Academy, Seoul 01805, Korea
3   ICT Polytech Institute of Korea, Gyeonggi 12777, Korea; iolapleader@gmail.com
*   Correspondence: parky@mnd.go.kr

**Abstract:** The Korean peninsula is under increasing threat of electromagnetic pulses (EMPs) from neighboring countries; EMP protection facilities are an essential means of ensuring the operational readiness of the military. However, existing EMP protection facilities are manufactured as fixed-weight structures, which limit the mobility of military operations and lead to the misconception of EMP protection as something only required for higher command. The current military and official EMP protection standards require only a uniform shielding effectiveness of 80 dB. Therefore, this study aims to differentiate the existing uniform level of shielding effectiveness of 80 dB into 80 dB, 60 dB, 40 dB, etc. Further, it seeks to derive the factors to be considered when applying various methods, such as shielding rooms, shielding racks, site redundancy, spare equipment, and portable lightweight protective tents, for recovery of failure, instead of the existing protection facilities that rely on shielded rooms by the Delphi analysis. Then, the applicability of lightweight EMP protection is determined after selecting lightweight materials to build a facility. The electromagnetic shielding performance of 21 types of materials was measured in the 30 MHz–1.5 GHz frequency band using ASTM-D-4935-10. The results showed the possibility of developing a lightweight EMP shielding facility, which would save approximately 316,386 tons of concrete, reducing the $CO_2$ emissions by approximately 9,972,489 tons. Assuming that the Korean carbon transaction price is USD 50/ton $CO_2$, the savings are equivalent to USD 49,862,435.

**Keywords:** electromagnetic pulse (EMP); shielding effectiveness (SE); EMP tent; METT+TC; sustainable military design; $CO_2$ emission; required EMP protection level

## 1. Introduction

### 1.1. Background

An electromagnetic pulse (EMP) is an electromagnetic shock wave with strong energy, which is a powerful weapon capable of destroying or jamming electric and electronic devices and systems simultaneously. EMP technology was developed in the 1960s as a nuclear explosion experiment conducted by the US and the Soviet Union; it revealed that electromagnetic waves could neutralize a wide range of electrical and electronic devices [1,2]. In July 1962, the US conducted a nuclear test (1.4 million tons of TNT) at an altitude of 400 km above Johnston Island in the Pacific under the project name Starfish Prime. This experiment damaged the observation equipment approximately 800 km away from the explosion site and the US military electronic communications surveillance and command system 1300 km away. Moreover, in Hawaii, which was approximately 1500 km away, electronic devices, such as traffic lights, radios, televisions, and electric fuses, were paralyzed for 30 min [3,4]. In South Korea, there are increasing concerns about EMPs due to nuclear and missile threats by North Korea. Because North Korea appears to focus on developing asymmetric strategic weapons, as in the fifth nuclear test and the caterpillar mobile launch vehicle test in September 2016, there is a high possibility that

EMP attacks may be carried out to destroy infrastructure such as power, communication, and finance, rather than as a nuclear provocation to claim casualties [5,6]. During a nuclear explosion 50 km above the armistice line, all of South Korea would be affected by an EMP. In addition to the EMP caused by nuclear attacks, there is also an increasing risk of small non-nuclear EMP weapons that can threaten infrastructure by intentionally generating EMPs in major information and communication infrastructure. Recently, equipment capable of generating an EMP, even with low power, has been miniaturized so that it can be carried by individuals. Such features enable individuals to easily execute EMP attacks anytime and anywhere, thereby causing continued threats of local terrorism, such as the invasion of major information and communications infrastructure, and intentionally generating EMPs to destroy electrical and electronic systems [7,8].

To cope with such threats, the South Korean government has established protective measures against EMPs for major national infrastructure in public and private sectors and is building EMP protection facilities for the military in preparation for EMP attacks [9–12]. However, the construction of EMP protection facilities for all national infrastructure would incur a significant cost. Currently, in most countries the US Department of Defense's EMP protection technology and test method standards (MIL-STD-188-125, etc.) are applied mutatis mutandis to EMP protection, but these standards are suitable for fixed facilities and required for a shielding effectiveness of 80 dB at 1 GHz [10]. These standards propose the building of an EMP protection facility by surrounding all systems with a metal enclosure; however, these do not consider the shielding and attenuation properties of the general buildings or underground facilities as they are the EMP protection standards for ground-based C4I (Command, Control, Communication, Computer, Intelligence) facilities. The protection facilities built to the current standards require a huge amount of concrete, rebar, and steel plates. However, reducing the use of concrete and iron in construction projects is crucial in terms of sustainability awareness and green planning [13]. According to the International Energy Agency and United Nations (UN) Environment Programme, 36% of the global energy is consumed by building construction and operations, accounting for 40% of energy-related carbon dioxide ($CO_2$) emissions in 2017 [14]. According to the research results of Pacheco-Torgal et al., 65% of building greenhouse gas (GHG) emissions are caused by the use of concrete and rebar, and 40% of $CO_2$ emissions are caused by the use of concrete [15]. In particular, the mean embodied carbon dioxide ($ECO_2$) of a building is 340 kg $CO_2/m^2$, accounting for approximately 60% of the total structure [16]. This means that reducing the $ECO_2$ of the structure has the same effect as reducing GHG emissions. Moreover, it suggests that reducing the use of concrete and rebar in construction projects is crucial in terms of carbon emissions [17–21]. Reducing carbon emissions in the construction of military facilities, which accounts for a large proportion of major construction projects in South Korea, is a policy stance that must be pursued to contribute to the green growth policy of the South Korean government at the defense level [22]. In addition, the lightweight EMP shielding facility can provide military advantages, such as rapid maneuvering and operational flexibility for successful operations.

Therefore, this study aims to develop a lightweight EMP shielding method that can guarantee economic and temporal efficiencies compared to the existing EMP protection method by applying differential EMP shielding effectiveness. The factors to be considered when determining the EMP protection level and the required EMP shielding effectiveness were derived through Delphi analysis. Through this, it is determined whether or not it is possible to obtain a military lightweight EMP shielding facility using various materials with a shielding effectiveness of 80 dB or less. Further, it aims to analyze the feasibility of developing an EMP shielding tent by performing a comparative verification and analysis of lightweight EMP shielding materials that can be applied to major national infrastructure, including the military. The materials verified by this study will ultimately be used in the development of a deployable tactical EMP tent for the South Korean military, specifically for ground forces who are required to move as the battlefield expands.

*1.2. Objectives and Scope*

This study is aimed at comparing and verifying EMP shielding materials that are applicable to military and national infrastructure. Accordingly, the factors for determining the EMP protection level were derived by performing Delphi analysis. The applicable EMP shielding materials were selected using the factors determined by Delphi analysis, and the shielding effectiveness of the selected materials against EMPs from 30 MHz to 1.5 GHz was measured using the ASTM-4935-10 test, which is the standard for EMP shielding performance. Using the results obtained by this method, the $CO_2$ emission and cost reduction effects of the lightweight EMP protection facility to be built in the future were analyzed.

## 2. Conventional EMP Shielding Method

Currently applicable MIL-STD-188-125 standards stipulate EMP protection performance for protection facilities up to 80 dB in the 10 kHz–1 GHz frequency bands. To achieve this performance, the materials commonly used for building protection facilities include plywood attached to a thin sheet of galvanized steel on one or both sides or particleboard panels (majorly used for prefabricated EMP shielding rooms) and steel support structures welded with steel plates. In general, a hard metal surface thicker than 0.01 in (0.254 mm) provides good EMP shielding effectiveness performance. Figure 1 shows an EMP shielding room commonly used in the military.

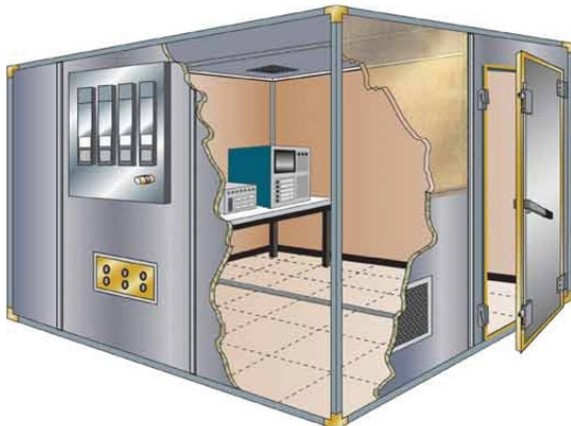

**Figure 1.** Conventional electromagnetic pulse (EMP) shielding method used for military command and control rooms [23].

There are three primary methods of constructing an EMP shielding room using the abovementioned shielding materials. The first is the modular panel type, which is the most commonly used electromagnetic shielding method. This type is a clamp-up system that uses galvanized steel plates joined together by a framework made of plated steel on both sides of the wooden plate. The galvanized panel system is the most commonly used method as it is readily available from many suppliers and provides good performance over a wide frequency range. The second is the modular PAN-type electromagnetic shielding method, which is a configuration that takes advantage of the modular electromagnetic shielding room while overcoming the disadvantage of shielding performance. The modular PAN type uses a steel plate made by cutting and bending a single plate. The size of the PAN is not standardized, and it is designed according to the desired specifications. It is usually assembled using bolts and nuts, and the structure is closely attached with a gasket. The advantages of this method include a light weight, excellent constructability, excellent load resistance and vibration resistance, easy mobility and installation, and short construction time. The third method is the well-established welding-type electromagnetic shielding method, which is used when high-performance electromagnetic wave shielding is required for a long time. In general, the welded EMP shielding room lasts 30 years

with a shielding performance of 120 dB against electric fields and plane waves. The fundamental drawbacks are that the construction cost is high, and it is impossible to move the structure after installation. Therefore, careful planning is required to provide the facility with optimal shielding performance. In terms of the cost and time required to build a new protection facility and changing the existing facility to a protection facility, there is a need for research on the construction of EMP protection facilities using lightweight electromagnetic shielding materials. To meet the level of protection required by the standards, most of the currently established protection facilities are built by welding, without considering the electromagnetic shielding properties of buildings and underground facilities.

Therefore, considering the electromagnetic shielding properties of existing buildings and underground facilities when constructing EMP protection facilities, the desired electromagnetic shielding performance may be achieved with lightweight protective materials, such as electromagnetic shielding fabric, sheets, wallpaper, and paint, without the need to surround all systems with metal.

## 3. Development of Lightweight EMP Protection Decision Factors as a Sustainable Approach

To use lightweight materials in EMP protection facilities using existing heavy structures, it is necessary to establish a differential protection level system that considers the resistance of the equipment, ease of maintenance and replacement, and maintenance time, instead of the existing uniform level of 80 dB. For example, various levels of EMP protection, such as 80 dB, 60 dB, and 40 dB, enable a more flexible response to EMP threats. Accordingly, a survey was conducted to differentiate the EMP protection levels and protection measures. A panel of 21 civilian and military experts (seven civilian experts, seven government officials, and seven servicemen) was formed to obtain opinions on differentiating an existing uniform level of 80 dB into various levels of 80 dB, 60 dB, and 40 dB, as well as on applying various shielding measures, including shielding rooms, shielding racks, site redundancy, spare equipment, and portable lightweight protective tents for failure recovery, instead of the conventional method that relies on shielding rooms using the Delphi technique. From the Delphi analysis, a statistically significant agreement was confirmed for applying differential EMP protection levels of 80 dB, 60 dB, and 40 dB presented by the International Electrotechnical Commission (IEC). To determine the factors to be considered for the construction of lightweight EMP shielding facilities, factors to be considered in the setup of EMP protection methods were obtained using the first questionnaire (open questions) and the second to fourth questionnaires (closed questions). In particular, the Shapiro–Wilk normality test was performed to quantify the consensus of the panel between each of the second to fourth questionnaires. Thereafter, factor analysis [24–26] was performed to analyze the commonality of considerations for each factor. Consequently, although not intended, it was summarized as the military tactical factors (METT + TC): enemy, mission, terrain, time, and troops. Then, the key factors for EMP protection in the future battlefield environment from the innovation school and innovation evaluation conducted by the Korea Army Research Center for Future and Innovation (KARCFI) were extracted as the keywords for vertical and horizontal integration. Therefore, the six major factors to consider when selecting an EMP protection method were identified: wartime and peacetime missions; omnidirectional threats; the stability and resilience of troops; geology and weather; threat detection, alert, reaction, and recovery time; and the military–private combined factor. As shown in Table 1, factors to be considered were derived to ensure the required performance while reducing the weight to provide flexibility in military operations, considering the various battlefield environments and weapon systems in the future.

**Table 1.** Factors considered during the EMP protection method selection.

| Classification | Considerations for the Protection Standard of Military Facilities | |
| :---: | :---: | :---: |
| | **Factors** | **Detailed Items** |
| M (Mission) | Wartime/peacetime mission (4 items) | Peacetime mission |
| | | Wartime mission |
| | | Importance of mission |
| | | Impact of equipment damage on the mission |
| E (Threat of the enemy) | Omnidirectional threat (4 items) | Present threat (tactics, weapons, and activity of the enemy) |
| | | Potential threat |
| | | Transnational threat |
| | | Non-military threat |
| T (Available troops) | Stability and resilience of troops (7 items) | Type of troops |
| | | Location and capacity of protective facility |
| | | Equipment vulnerable to EMP attacks |
| | | Spare equipment |
| | | Skilled technicians for recovery |
| | | Damage recovery capacity (each unit and higher command) |
| | | Threat response capacity |
| T (Terrain and weather) | Geology and weather (3 items) | Natural EMP attenuation |
| | | Man-made EMP attenuation |
| | | Ground and underground stations |
| T (Available time) | Threat detection, alert, reaction, and recovery time (5 items) | Threat detection time |
| | | Threat alert time |
| | | Time taken to recover damaged equipment |
| | | Time taken to replace damaged equipment |
| | | Time for neutering threat |
| C (Civil factor) | Military–private combined factor (4 items) | Impact of civilian damage on the country |
| | | Impact of civilian damage on the military |
| | | Importance of facility |
| | | Time taken to recover damage |

## 4. Lightweight EMP Shielding Material Test

### 4.1. Review of Lightweight EMP Shielding Material

A high electromagnetic shielding effect is associated with an excellent level of protection against electromagnetic waves. Numerous metals with excellent electrical properties are being used as representative electromagnetic shielding materials, but they are heavy, easily corroded, and difficult to process. Recently, to compensate for these shortcomings, composite materials with polymers, which are lightweight, environmentally resistant, and possess high productivity, have emerged. When fiber materials and structures contain electromagnetic properties, fiber materials can be used to shield electromagnetic waves. In general, fiber materials are known to have excellent electrical insulating properties. Most fibers are made of natural or synthetic polymers with significantly high electrical resistance. However, the electrical conductivity of fiber materials is required in certain applications, such as electric heating, protection from electromagnetic waves, and signal transmission. Therefore, many attempts have been made to manufacture conductive fibers. The types

of commercial electromagnetic wave shielding materials related to construction that are currently available can be classified into four broad categories: fabric, wallpaper, sheet, and tape. It also includes new materials that have recently been developed. Most of these materials are not composed of a single raw material, but are made by mixing with different metal components to maximize the shielding performance against electromagnetic waves. Such a component mixing method increases the electromagnetic shielding performance and durability and corrosion resistance of the shielding material. In particular, for metal–polymer composites, metal fibers with a large aspect ratio have attracted attention as they can demonstrate excellent electromagnetic shielding characteristics with increased conductivity and are lightweight because the high aspect ratio enlarges the conductive path, even at low content.

Conductive fibers are suitable for electromagnetic shielding applications with their excellent electromagnetic shielding properties, offering many advantages, such as a light weight, elasticity, porosity, air permeability, corrosion resistance, and low cost, compared to metal foil or a metal grid. Fibers with an electromagnetic shielding effect have a wide range of applications. Conductive fibers are made by mixing metal with raw materials using various methods. For example, coating the outer part of the yarn with metal components, such as silver (Ag) and copper (Cu), on the polyester fiber can provide electromagnetic shielding properties. It is produced by electroless plating, which is a method of performing Ni plating prior to Cu plating and performing Ni plating on it to strengthen the adhesion of the plated layers. Another electromagnetic shielding method for fibers is to use magnetic metal hollow fibers. This method performs electroless plating with a magnetic metal on the surface of the microfiber and removes the microfiber using the heat treatment process to produce a hollow fiber. This creates a hollow fiber structure to make it more lightweight as filler and controls the structure and phase of the magnetic metal layer for increased crystallinity and compactness of the structure. In addition, it demonstrates appropriate electromagnetic shielding performance in the frequency bands of 30 MHz and 10 GHz, and even in the ultra-high frequency band. Figure 2 shows an SEM image of the FeCo magnetic metal hollow fiber manufactured by the electroless plating and high-temperature heat treatment.

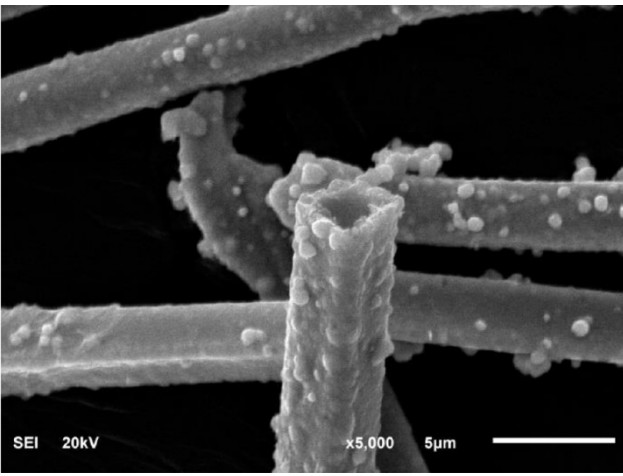

**Figure 2.** SEM image of the FeCo magnetic metal hollow fiber [27].

Before the dangers of electromagnetic waves and their effects on the human body were widely popular, electromagnetic shielding sheets were primarily used in hospitals to secure the visibility of the window to spatially separate the magnetic resonance imaging (MRI) scanning device from the controller, thereby preventing the effects of the electromagnetic waves generated during MRI imaging on the controller. Because the dangers of electromagnetic waves are now widely known, electromagnetic shielding sheets have been actively researched and developed to protect devices that are extensively used by humans

and that generate electromagnetic waves, such as mobile phones, TVs, PC monitors, and tablet PCs. The electromagnetic shielding sheet comprising a thin layer is selected and layered into a certain thickness to be attached by applying a conductive adhesive. With the development of technology, the electromagnetic shielding sheet becomes thinner, more flexible, and more durable. Currently, thin films are formed by implementing the electroless plating technique of a plating company. However, thin film deposition with metal using the sputtering method can form a thin and uniform film compared to using the electroless plating technique, requiring a simpler device. Therefore, there have been various attempts to realize its commercialization.

Electromagnetic shielding paint, which is one of the methods used to prevent damage from electromagnetic waves, is a product that is obtained by dispersing metal fillers of Ni, Cu, and Ag on the surface of resin components, such as acrylic and urethane, and it has many advantages. In other words, because the existing production line can be used as is, no additional investment is required, and product coating can be easily performed with only spray. It can be performed on all plastics by painting over the paint to reinforce the surface of the base material and is very cheap compared with other electromagnetic shielding methods. However, the disadvantages include that finishing is difficult, a low resistance compared with the plating and disposition methods, the film can be peeled off, there is inadequate uniformity without a robot sprayer as the coated film is rather thick (approximately 45 μm), and the long-term shielding effect may be inferior owing to oxidation of the filler, such as Cu.

### 4.2. Selection of Test Materials

In this study, there are practical limitations when purchasing all electromagnetic shielding materials that are available at home and abroad. To solve this problem, the products that are suitable for the experiment were selected as the evaluation criteria based on the price, shielding performance proposed by the manufacturer, shipping time, shipping area, material composition, and manufacturing method. Accordingly, the products were selected, as listed in Table 2.

**Table 2.** Shielding material selected.

| Classification | Manufacturer | Product Number |
|---|---|---|
| Shielding fabric or wallpaper (13 types) | Samgang tech (Seoul, Korea) | SGF-D130 (Fabric) |
| | Samgang tech | SGF-D150 (Fabric) |
| | Samgang tech | SGF-WD270 (Fabric) |
| | A-Jin Electron (Busan, Korea) | W-290-PCN (Fabric) |
| | Hana Elecom (Incheon, Korea) | CFT-235-FR-NH (Wallpaper) |
| | Hana Elecom | CFT-290-FR-NH (Wallpaper) |
| | Holland Shielding (Dordrecht, The Netherlands) | Systems BV 4711 series (Fabric) |
| | Less EMF Inc. (Latham, NY, USA) | Stick E Shield (Wallpaper) |
| | Less EMF Inc. | COBALTEX (Fabric) |
| | Less EMF Inc. | NICKEL/COPPER RIPSTOP FABRIC (Fabric) |
| | Less EMF Inc. | PURE COPPER POLYESTER TAFFETA (Fabric) |
| | Less EMF Inc. | SILVER MESH FABRIC (Fabric) |
| | Y-Shield (Rotthofer, Germany) | YCF-60-100 (Wallpaper) |
| Shielding film (5 types) | EMCPRO (Incheon, Korea) | SF2209 |
| | Whil KOR (Irvine, CA, USA) | WT 70 MNT |
| | ShieldGreen (Gyeonggi-do, Korea) | SGWF26 |
| | Less EMF Inc. | Scotch Tint |
| | Less EMF Inc. | Scotch Tint Super |
| Tape (3 types) | E-Song EMC | Metal foil tape, copper |
| | E-Song EMC (Seoul, Korea) | Metal foil tape, aluminum |
| | Holland Shielding | BV 3212 series |

*4.3. Shielding Effectiveness (SE) Test Results*

The ASTM-D-4935-10 method was implemented to evaluate the shielding effectiveness of the selected material. This measurement method was applied to measure the electromagnetic shielding effect of a planar material due to reflection and absorption by a planar material under the condition that the plane waves of a far field are incident. The frequency required by the standard ranges from 30 MHz to 1.5 GHz. The equipment required for the measurement comprises a signal generator, attenuation, specimen holder, and receiver. However, when a network analyzer is used, signal transmission and reception can be performed with only one device, allowing simple measurement. The signal generator should be able to generate a sine wave signal in the required measurement frequency band, and it should have an output impedance characteristic of 50 $\Omega$ for impedance matching with the specimen holder. A tracking signal generator having the same characteristics can also be used. The receiver should have an input impedance of 50 $\Omega$ and should be able to receive a signal in the frequency range generated by the signal generator; further, it requires a wide operating range that can sufficiently measure the characteristics of the test specimen. The operating range refers to the difference between the maximum and minimum signal levels that can be measured in a measurement system. The commonly used receivers include a spectrum analyzer or an electric field strength meter. A bidirectional coupler can also be used to observe the change in incident power due to the energy reflected by the specimen.

The specimen holder is a structure of a circular coaxial line extending from both ends through an inclined surface. The impedance between the ends should be maintained at 50 $\pm$ 0.5 $\Omega$ within the measurement frequency band. In addition, there should be a pair of flanges in the center of the support, and the EUT should be supported by these flanges for a capacitive coupling between the insulating material and displacement current. The test specimens consist of reference and load specimens, and the thickness and physical or electrical characteristics of these specimens should be the same. Specimens should be stored in a constant temperature/humidity environment specified by the standards, and any homogeneous or heterogeneous, single or multilayer conductivity and insulation can be measured. Consequently, the measured value of the shielding effect of a heterogeneous material depends on the installation position and orientation, and the reproducibility of the measurement result is lower than that of a homogeneous material. Moreover, the accuracy of the measurement result decreases as the specimen thickness increases. Figure 3 shows the ASTM-D-4935-10 test setup, and Table 3 presents the electromagnetic shielding efficiency by frequency band.

To measure the shielding performance of the material itself using the ASTM method, specimens for each material should be manufactured in a certain standard. The manufacturing was performed by a company that specialized in laser cutting to produce specimens with accurate dimensions. Among the films, the SF2209 model of EMC PRO exhibited the highest shielding performance of 35.0 dB in the 30-MHz frequency band. It also demonstrated an overall high performance in other frequency bands. Among the fiber and textile materials, the model of Holland Shielding exhibited the highest shielding performance of 87.1 dB in the 50-MHz frequency band. Among the wallpapers, the Stick E Shield model of Less EMF demonstrated the most excellent shielding performance of 71.3 dB in the 30-MHz frequency band. Among the shielding tapes, the metal foil tape (Cu) model of Song EMC showed the best shielding performance at 99.2 dB in the 1.5-GHz frequency band.

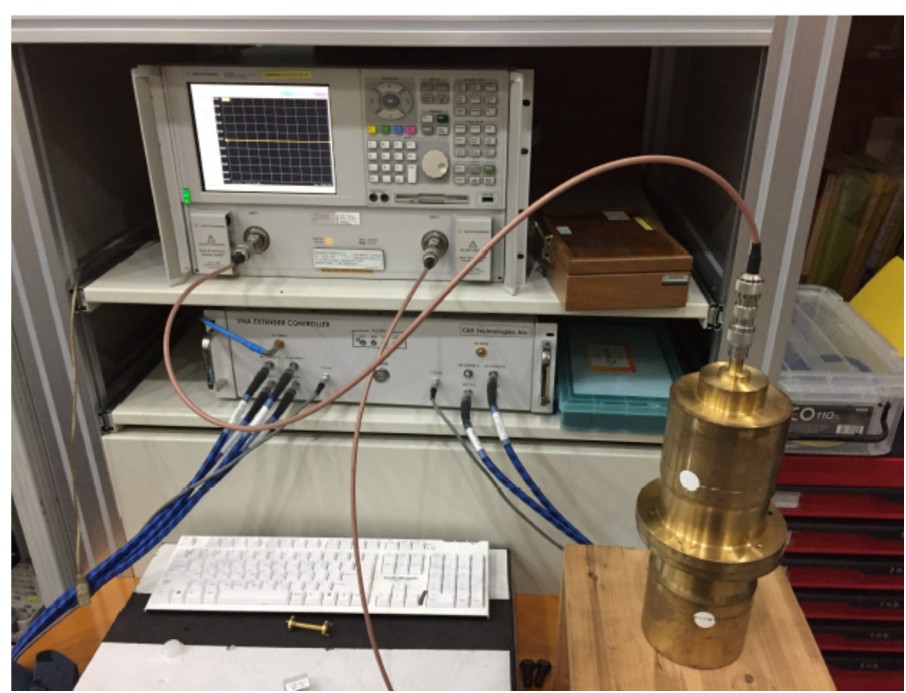

**Figure 3.** ASTM-D-4935-10 test setup.

**Table 3.** Shielding effectiveness test results.

| Product Number | Frequency (MHz) | | | | | |
|---|---|---|---|---|---|---|
| | 30 | 50 | 100 | 400 | 1000 | 1500 |
| SGF-D130 | 68.1 | 68.1 | 66.0 | 71.2 | 75.1 | 76.8 |
| SGF-D150 | 58.3 | 59.1 | 59.5 | 61.1 | 62.6 | 63.2 |
| SGF-WD270 | 66.1 | 65.0 | 66.1 | 68.9 | 74.5 | 74.6 |
| W-290-PCN | 64.3 | 64.5 | 65.4 | 68.4 | 70.3 | 71.5 |
| CFT-235-FR-NH | 66.6 | 66.2 | 64.1 | 65.2 | 67.8 | 67.9 |
| CFT-290-FR-NH | 65.7 | 65.1 | 63.0 | 63.8 | 66.2 | 67.2 |
| Systems BV 4711 series | 71.9 | 87.1 | 72.8 | 71.8 | 76.0 | 75.4 |
| Stick E Shield | 78.0 | 75.3 | 72.6 | 73.7 | 74.0 | 72.0 |
| COBALTEX | 68.6 | 77.0 | 71.6 | 73.7 | 77.9 | 80.0 |
| NICKEL/COPPER RIPSTOP FABRIC | 77.2 | 70.2 | 73.6 | 75.1 | 78.3 | 77.6 |
| PURE COPPER POLYESTER TAFFETA | 70.8 | 74.2 | 74.3 | 77.7 | 79.4 | 78.5 |
| SILVER MESH FABRIC | 32.9 | 32. | 31.8 | 32.0 | 34.6 | 37.1 |
| YCF-60-100 | 71.3 | 66.5 | 67.3 | 64.2 | 60.9 | 58.9 |
| SF2209 | 35.0 | 34.3 | 34.4 | 34.5 | 34.3 | 34.8 |
| WT 70 MNT | 22.6 | 22.2 | 22.2 | 22.6 | 23.1 | 23.8 |
| SGWF26 | 23.9 | 23.4 | 23.5 | 23.8 | 24.2 | 24.8 |
| Scotch Tint | 32.5 | 32.8 | 33.2 | 33.4 | 33.5 | 34.0 |
| Scotch Tint Super | 23.2 | 21.7 | 21.0 | 21.0 | 21.4 | 22.0 |
| Metal foil tape, copper | 86.9 | 72.3 | 80.8 | 83.1 | 82.3 | 73.8 |
| Metal foil tape, aluminum | 90.5 | 79.9 | 85.6 | 84.3 | 86.1 | 99.2 |
| BV 3212 series | 79.5 | 74.5 | 90.1 | 92.3 | 84.5 | 88.0 |

*4.4. CO$_2$ Emission Reduction Effects*

According to the research results to date, experts agreed that the current uniform 80-dB shielding standard of the Korean military should be differentiated into various levels and methods. In addition, the need to develop a more lightweight and flexible shielding facility than the conventional heavy EMP protection facility was confirmed. By applying the lightweight shielding facility and the differentiated shielding level system, it was possible to produce a deployable tactical EMP tent, which is more useful in military operations. Further, the application of such a lightweight structure to military facilities comply with the military sustainability policy. In addition, the application of lightweight protection facilities instead of building new EMP protection facilities would save a significant amount of concrete. Table 4 lists the amount of CO$_2$ emissions savings realized for concrete upon replacing 20 EMP protection facilities by lightweight facilities. Assuming that the unit CO$_2$ emissions of ready-mixed concrete is 3.152 ton CO$_2$/ton [28], the CO$_2$ emissions generated by the construction of EMP protection facilities can be reduced by approximately 997,249 tons. Assuming that the Korean carbon transaction price is USD 50/ton CO$_2$ [29], it is equivalent to USD 49,862,435. Moreover, the lightweight EMP protection facility enables the military and private sectors to easily respond to EMP threats. Therefore, it has the advantage of flexible application to changes in the surrounding environment. This can contribute to the national defense of South Korea and the policy stance of green growth.

**Table 4.** Calculation of the CO$_2$ emission reduction effect.

| Description | Quantity (ton) | Unit CO$_2$ Emission (ton CO$_2$/ton) | Amount (ton CO$_2$) |
| --- | --- | --- | --- |
| Conventional EMP room (A) | 340,200 | 3.152 | 1,072,310.4 |
| Lightweight EMP room (B) | 23,814 | 3.152 | 75,061.7 |
| Reduction effect (A–B) | 316,386 | | 997,248.7 |

## 5. Conclusions

South Korea is under threat of nuclear tests and missile provocations by North Korea, while the technological trend in major defense weapons is shifting to electronic warfare. Therefore, EMP threats are increasing day-by-day, and South Korea is also establishing EMP protection measures for major national infrastructure with an increasing awareness of EMP threats around the world. However, because of differences in the scale, size, and characteristics of the expected damage of each major facility, the building of EMP protection facilities for all infrastructure is subject to practical limitations, such as enormous costs and time. To solve this problem, there is a constant need to develop a more lightweight protection facility than the existing EMP protection facility. However, up to the present, there are no facilities that have implemented such lightweight protection technology. Therefore, this study aimed to verify the performance of lightweight electromagnetic shielding materials and to review the feasibility of the development of a deployable tactical EMP tent.

Accordingly, the existing uniform level of the shielding effectiveness of 80 dB was differentiated into 80 dB, 60 dB, 40 dB, etc., and it derived the factors to be considered for applying various methods, such as shielding rooms, shielding racks, site redundancy, spare equipment, and portable lightweight protective tents for failure recovery, instead of the existing protection facilities that rely on shielded rooms by the Delphi analysis. Consequently, the six factors to consider when selecting an EMP protection method were identified: wartime and peacetime missions; omnidirectional threats; the stability and resilience of troops; geology and weather; threat detection, alert, reaction, and recovery time; and a military–private combined factor.

Lightweight materials were selected for constructing the lightweight EMP protection facility. First, the technical trends in the methods for measuring electromagnetic shielding performance and the technological trends in materials were analyzed and organized according to domestic and foreign literature and various papers. Based on these trend

data, electromagnetic shielding materials that are suitable for weight reduction among commercially available products were investigated and catalogued. The investigated materials primarily included fibers and textile materials, films, wallpaper, and paint suitable for weight reduction. Because there are limitations in terms of time and cost for purchasing and testing all the investigated electromagnetic shielding materials, the products suitable for the actual testing were selected according to specific criteria. According to the selection criteria, 21 materials, including nine types of fibers, four types of wallpapers, five types of films, and three types of tapes, were selected. The shielding performance of each material was measured in the frequency band of 30 MHz–1.5 GHz using ASTM-D-4935-10. Through this, the development of a lightweight EMP shielding facility was found to be possible. Replacing the EMP protection facility required by the military with a lightweight EMP protection facility would save approximately 316,386 tons of concrete, reducing $CO_2$ emissions by approximately 997,248.7 tons. Assuming that the Korean carbon transaction price is USD 50/ton $CO_2$, it is equivalent to USD 49,862,435.

Although this study confirmed the possibility of developing a deployable tactical EMP tent for the military, regardless of how high the shielding effect of the fibers, films, and wallpaper is against electromagnetic waves, the shielding performance and durability would still be lower than those of metal steel plates. In other words, fibers, films, and wallpaper are sensitive to light (such as sunlight), changes in temperature and humidity, external impact, and corrosion. Therefore, they may require constructional reinforcements, such as coating with transparent paint or construction to prevent the material from being exposed to the outside, as well as additional measures to enhance the durability and electromagnetic shielding performance by analyzing the shielding performance of each material.

**Author Contributions:** Conceptualization, Y.-J.P. and K.K.; Methodology, K.K. and K.-R.M.; Software, K.-R.M.; Validation, Y.-J.P., K.K. and , K.-R.M.; Formal Analysis, K.K.; Investigation, K.-R.M.; Resources, Y.-J.P.; Data Curation, K.-R.M. and K.K.; Writing-Original Draft Preparation, K.K.; Writing-Review & Editing, Y.-J.P.; Visualization, K.-R.M.; Supervision, Y.-J.P.; Project Administration, K.-R.M. and Y.-J.P.; Funding Acquisition, Y.-J.P. All authors have read and agreed to the published version of the manuscript.

**Funding:** This research was supported by a grant (20SCIP-B146646-03) from the Korea Agency for Infrastructure Technology Advancement.

**Acknowledgments:** This work was supported by a research fund of the Korea Agency for Infrastructure Technology Advancement. The ROKA Nuclear·WMD Protection Research Center at the Korea Military Academy is gratefully acknowledged for providing the support that made this study possible.

**Conflicts of Interest:** The authors declare no conflict of interest.

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
