# Peer review of "Feasibility Study on the Development of a Deployable Tactical EMP Tent for a Sustainable Military Facility"

_sustainability, doi:10.3390/su13010016_

Round 1
Reviewer 1 Report
A very well organized work, easy to follow. The research methods are clearly described and the results and conclusions support the research approach.
My only recommendation is to improve bibliographic sources, so that the information is better perceived in terms of background.
Author Response
First, the authors would like to acknowledge and thank all reviewers for providing invaluable and constructive review comments in relation to this article. These comments are sincerely appreciated and have clearly made a positive impact on the quality of the paper. Based on the reviewers’ suggestions, changes were done to the paper to make it hopefully clearer and more understandable.
Response to Reviewer 1
- A very well organized work, easy to follow. The research methods are clearly described and the results and conclusions support the research approach. My only recommendation is to improve bibliographic sources, so that the information is better perceived in terms of background.
Re: The authors appreciate your comment. Several references were added to reinforce the background logic.
Reviewer 2 Report
Thank you for the opportunity to review your paper. I agree that EMPs present a significant future threat for militaries and governments around the world, and there is tremendous value in figuring out how to cheaply and quickly construct EMP-resistant facilities.
The objective of this paper was to measure the capability of various EMP shielding materials. The authors discuss "deriving the factors to be considered when deriving the protection level and the required protection level reflecting various factors" but there is almost no information provided about the details of the Delphi study.
I believe this paper was submitted to sustainability because the authors claim by (1) lowering protection requirements, and (2) using thinner/lighter materials, a significant reduction in concrete use could be achieved.
The sustainability aspects of this paper seemed to be a late edition and not the main focus of the paper. Far more background is given to discuss the history and development of lightweight EMP shielding material than considering design changes and your associated savings claims. Additionally, you claim a cost savings associated with concrete reduction, but you don't compare the additional cost associated with potentially using more expensive materials.
You only claim the benefits of reduced concrete, including tons, emissions, and emission cost-avoidance. You do not discuss the additional costs, emissions, sustainability impacts, etc. of the selected shielding materials. Therefore, the sustainability picture is incomplete.
I have a few other minor points:
1) please adjust the number of significant figures
2) add a citation re: Starfish Prime
3) line 66 - please clarify what you mean by "protection facilities of heavy structures require"
4) line 83-86 - This is the key objective of your paper, but the length makes is confusing. I suggest breaking this into two sentences and making more readable.
5) line 94 - determinants for determining is redundant and confusing
6) Table 1 - don't break tables across pages
7) Additional detail needs to be provided on the Delphi study. What questions were asked? What were the results? Your entire basis for calculating the reduced material requirements is based on the Delphi study, but there is almost no explanation of the process or results.
8) line 176 - "shielding" not "sheilding"
9) line 181 should have "lightweight"
10) line 189 - you claim four categories but only list three
11) most of section 4.1 is not needed
12) Table 2 - "classification" shouldn't break lines
Author Response
First, the authors would like to acknowledge and thank all reviewers for providing invaluable and constructive review comments in relation to this article. These comments are sincerely appreciated and have clearly made a positive impact on the quality of the paper. Based on the reviewers’ suggestions, changes were done to the paper to make it hopefully clearer and more understandable.
Response to Reviewer 2
1. Thank you for the opportunity to review your paper. I agree that EMPs present a significant future threat for militaries and governments around the world, and there is tremendous value in figuring out how to cheaply and quickly construct EMP-resistant facilities. The objective of this paper was to measure the capability of various EMP shielding materials. The authors discuss "deriving the factors to be considered when deriving the protection level and the required protection level reflecting various factors" but there is almost no information provided about the details of the Delphi study.
Re: The EMP protection standards are suitable for fixed facilities and required for shielding effectiveness of 80 dB at 1 GHz. These standards propose the building of an EMP protection facility by surrounding all systems with metal plates; however, these do not consider the shielding and attenuation properties of general buildings or underground facilities as they are the EMP protection standards for ground-based C4I (Command, Control, Communication, Computer, Intelligence) facilities. Through Delphi study, it is determined whether or not it is possible to obtain a military lightweight EMP shielding facility various materials with the shielding effectiveness of 80 dB or less. This paper has been revised to help readers understand. Please refer to the revised manuscript in line 152-156.
2. I believe this paper was submitted to sustainability because the authors claim by (1) lowering protection requirements, and (2) using thinner/lighter materials, a significant reduction in concrete use could be achieved. The sustainability aspects of this paper seemed to be a late edition and not the main focus of the paper. Far more background is given to discuss the history and development of lightweight EMP shielding material than considering design changes and your associated savings claims. Additionally, you claim a cost savings associated with concrete reduction, but you don't compare the additional cost associated with potentially using more expensive materials. You only claim the benefits of reduced concrete, including tons, emissions, and emission cost-avoidance. You do not discuss the additional costs, emissions, sustainability impacts, etc. of the selected shielding materials. Therefore, the sustainability picture is incomplete.
Re: EMP protection in the current military uniformly requires 80dB of shielding effectiveness. This is the first study to use new materials with differential shielding effectiveness in the military. The new shielding materials are made from carbon and steel fibers, and evaluating the impact of these materials in term of a sustainability perspective will be a whole new study. Nevertheless, the authors believe that EMP protection using new materials is a very important matter from the viewpoint of military sustainability.
3. I have a few other minor points:
Re: All minor point you mentioned have been corrected. Please refer to the revised manuscript.